# Identification of Drugs Acting as Perpetrators in Common Drug Interactions in a Cohort of Geriatric Patients from Southern Italy and Analysis of the Gene Polymorphisms That Affect Their Interacting Potential

**DOI:** 10.3390/geriatrics8050084

**Published:** 2023-08-24

**Authors:** Mauro Cataldi, Camilla Celentano, Leonardo Bencivenga, Michele Arcopinto, Chiara Resnati, Annalaura Manes, Loreta Dodani, Lucia Comnes, Robert Vander Stichele, Dipak Kalra, Giuseppe Rengo, Francesco Giallauria, Ugo Trama, Nicola Ferrara, Antonio Cittadini, Maurizio Taglialatela

**Affiliations:** 1Department of Neuroscience, Reproductive Sciences and Dentistry, Federico II University of Naples, Via Sergio Pansini 5, 80131 Naples, Italy; camillacelentano5@gmail.com (C.C.); resnati.chiara@gmail.com (C.R.); annalaura.manes@gmail.com (A.M.); loretadodani@hotmail.com (L.D.); mtaglial@unina.it (M.T.); 2Department of Translational Medical Sciences, Federico II University of Naples, Via Sergio Pansini 5, 80131 Naples, Italy; leonardobencivenga@gmail.com (L.B.); michele.arcopinto@unina.it (M.A.); giuseppe.rengo@unina.it (G.R.); giallauriafrancesco@unina.it (F.G.); nicferra@unina.it (N.F.); antonio.cittadini@unina.it (A.C.); 3Gérontopôle de Toulouse, Institut du Vieillissement, CHU de Toulouse, Cité de la Santé, Place Lange, 31300 Toulouse, France; 4Datawizard, Via Salaria 719a, 00138 Rome, Italy; l.comnes@datawizard.it; 5Heymans Institute of Pharmacology, Ghent University, C. Heymanslaan 10, 9000 Ghent, Belgium; robert.vanderstichele@ugent.be (R.V.S.); dipak.kalra@i-hd.eu (D.K.); 6European Institute for Innovation through Health Data, c/o Department Medical Informatics and Statistics, Ghent University Hospital, C. Heymanslaan 10, 9000 Ghent, Belgium; 7Istituti Clinici Scientifici—ICS Maugeri S.p.A., Via Bagni Vecchi 1, 82037 Telese, Italy; 8General Directorate for Health Protection and Coordination of the Regional Health System, Regione Campania, Centro Direzionale Is. C3, 80132 Naples, Italy; ugo.trama@regione.campania.it

**Keywords:** cytochromes, drug interactions, medication review, polypharmacy, pharmacogenomics

## Abstract

Background: Pharmacogenomic factors affect the susceptibility to drug–drug interactions (DDI). We identified drug interaction perpetrators among the drugs prescribed to a cohort of 290 older adults and analysed the prevalence of gene polymorphisms that can increase their interacting potential. We also pinpointed clinical decision support systems (CDSSs) that incorporate pharmacogenomic factors in DDI risk evaluation. Methods: Perpetrator drugs were identified using the Drug Interactions Flockhart Table, the DRUGBANK website, and the Mayo Clinic Pharmacogenomics Association Table. Allelic variants affecting their activity were identified with the PharmVar, PharmGKB, dbSNP, ensembl and 1000 genome databases. Results: Amiodarone, amlodipine, atorvastatin, digoxin, esomperazole, omeprazole, pantoprazole, simvastatin and rosuvastatin were perpetrator drugs prescribed to >5% of our patients. Few allelic variants affecting their perpetrator activity showed a prevalence >2% in the European population: CYP3A4/5*22, *1G, *3, CYP2C9*2 and *3, CYP2C19*17 and *2, CYP2D6*4, *41, *5, *10 and *9 and SLC1B1*15 and *5. Few commercial CDSS include pharmacogenomic factors in DDI-risk evaluation and none of them was designed for use in older adults. Conclusions: We provided a list of the allelic variants influencing the activity of drug perpetrators in older adults which should be included in pharmacogenomics-oriented CDSSs to be used in geriatric medicine.

## 1. Introduction

Drug–drug interactions (DDI), i.e., changes in the effects of a drug caused by the concomitant administration of another drug [1], affect more than 25% of patients in the age range between 70 and 79 [2]. The high prevalence of DDIs in older adults can be easily explained by the fact that more than 65% of them are on polypharmacy (i.e., they take five or more pharmaceutical ingredients a day) [3,4], since the probability of developing DDIs increases exponentially with the number of prescribed drugs [5]. Usually, in DDIs, a first drug, known as the *perpetrator*, modifies the activity of a second drug, which is called the *victim* of the interaction. Sometimes, reciprocal interactions occur between drugs that are at the same time perpetrators and victims. Depending on the interactors involved, DDIs may either enhance or reduce the effects of the victim drugs causing, respectively, drug toxicity or therapeutic failure [6,7,8,9]. Therefore, to make pharmacological treatments safe and effective, individual therapies should be adjusted to prevent DDI occurrence by avoiding the prescription of dangerous combinations of potentially interacting drugs [10]. Physicians may be helped in this process by clinical decision support systems (CDSS) incorporating DDI checkers. Even better, therapy optimization can be obtained through *medication review*, a structured revision of pharmacological treatment performed by a team of pharmacologists and pharmacists, to identify drugs that are inappropriate, dangerous or that can be responsible for DDIs [11]. An important limitation of these interventions is that many of the identified DDIs do not have major clinical consequences, and this may cause “alert fatigue” in medical doctors with the consequence that even major DDI warnings and related recommendations are ignored [10,12]. It would be, therefore, highly desirable to introduce predictors that could help identify patients among those receiving potentially interacting drugs who are at highest risk of clinically relevant interactions and deserving the highest attention. The assessment of pharmacogenomic factors may help achieve this goal considering that the severity of DDIs could be affected by patient genetic background [13,14]. In fact, variants of genes involved in pharmacokinetics or pharmacodynamics may not only affect the efficacy of specific drugs [15,16], establishing the so-called drug–gene interactions (DGIs), but also influence the ability of specific drugs to act as perpetrators in DDIs. For instance, phase I metabolism of many of the drugs used in therapy depend on enzymes belonging to the cytochrome P450 (CYP450) superfamily, whose members CYP3A4/5, CYP2C9, CYP2C19, and CYP2D6, have largely different substrate specificity [17]. CYP450s are encoded by polymorphic enzymes and, therefore, in the general population, individuals can be identified who display enzymatic activity of one or more of these CYPs that is higher, slightly reduced or lower than normal [17]. They are called, respectively, ultrarapid metabolizers (UM), intermediate metabolizers (IM) and poor metabolizers (PM). Since their enzyme activity is lower than normal, PMs and, in some cases also IMs, are expected to be more susceptible to the effect of drugs that can further inhibit the already low CYP450 activity and, therefore, to display higher than normal exposure to drugs metabolized by these CYP450s and, possibly, develop toxicity. By contrast, UMs, who already have higher than normal enzyme activity, are expected to be more susceptible than normal individuals to drugs that induce CYP450 expression and, consequently, to display lower than normal levels and, possibly, therapeutic failure when exposed to drugs metabolized by the affected CYP450. Another key player in pharmacokinetic DDIs with a significant pharmacogenomic variability is the drug transporter SLCO1B1, which is involved in the elimination of important drugs such as statins or angiotensin receptor blockers (ARBs) [18]. SLCO1B1 is encoded by a polymorphic gene and people bearing its loss of function (LoF) variants are expected to be more susceptible than normal people to SLCO1B1 blockers, which further reduce the already low transporter activity and may, therefore, increase the plasma concentrations of SLCO1B1 substrates up to toxic levels. The individual pharmacogenotype should, therefore, be added to the list of the factors that can establish the severity of drug interactions, which also include the different individual exposure to environmental factors, biopharmaceutical considerations concerning the different preparations of the same active principles, individual differences in patient compliance to therapy and in the attending physician to the monitoring of early signs and symptoms of drug toxicity. The term drug–drug–gene interactions (DDGI) has been introduced to acknowledge the contribution of genetics in drug interactions [13,14]. Some evidence has been reported suggesting that pharmacogenetic testing combined with CDSS could prevent serious DGIs and, consequently, reduce hospitalizations, emergency department admissions, and outpatient visits in older adults on polypharmacy and alert fatigue in their doctors [19,20]. On the contrary, so far, only a few studies have investigated DDGIs in older adults and the impact of the prevalence of variants in key pharmacogenes on their occurrence [21,22]. Therefore, in the present study, we identified the drugs most frequently prescribed in a cohort of geriatric patients from Southern Italy and correlated to the prevalence of the main variants of the genes affecting disposition of the drugs most frequently prescribed in these subjects. In addition, given the role played by CDSS in the medication review process, and in consideration of the relevant role played by pharmacogenomics in identifying DDGIs, we also assessed whether tools commonly available for medication review incorporate DDGI checklists.

## 2. Materials and Methods

### 2.1. Study Design

To find gene variants that could enhance the susceptibility to common DDIs in older adults, we first identified the most prescribed drugs in a cohort of geriatric patients by means of a retrospective, observational analysis of prescription records. Then, we looked for potential DDI perpetrators among drugs prescribed to more than 5% of our population. Finally, we examined the allelic distribution of the genes influencing the pharmacokinetics of these potential DDI perpetrators focusing on those frequently occurring in Europe and, when available, specifically in Southern Italy.

### 2.2. Study Population

The population examined consisted of two groups of patients of both genders followed as part of the medication review program at the Federico II University Hospital of Naples, Italy: 1. Patients admitted to the Internal Medicine-Cardiac Rehabilitation ward, 2. Outpatients followed at the Geriatrics clinic. Inclusion criteria were age older than 65 years and treatment with at least 5 active pharmaceutical ingredients. Patients were excluded in the presence of at least one of the following conditions: artificial nutrition either enteral or parenteral, continuous intravenous drug therapy, KDIGO stage 5 chronic kidney disease, peritoneal dialysis or hemodialysis, Child-Pugh class C liver failure, chemotherapy for malignant tumors or immunosuppressant therapy for autoimmune diseases or organ transplantation. Ethical approval for the study was granted by the Ethics Committee of the Federico II University of Naples, ITALY (approval number 202/16).

### 2.3. Identification of Potential DDI Perpetrator Drugs

To identify potential DDI perpetrators among the drugs most frequently prescribed to the patients of our cohort, we matched the list of these drugs with those of the inhibitors and of the inducers of drug-metabolizing enzymes and transporters. More specifically. we focused on the cytochromes CYP3A4/5, CYP2C9, CYP2C19 and CYP2D6 and on SLC1B1 transporters since they are responsible for most of the DDIs commonly observed in the clinic [23]. The lists of the inhibitors and inducers of these enzymes and transporters were obtained, for CYPs, from the Drug Interactions Flockhart Table™ (freely downloadable at https://drug-interactions.medicine.iu.edu/MainTable.aspx, accessed on 27 January 2023) [24], the Mayo Clinic Pharmacogenomics Association Table (https://www.mayocliniclabs.com/it-mmfiles/Pharmacogenomic_Associations_Tables.pdf, accessed on 27 January 2023), the DRUGBANK online website (www.go.drugbank.com, accessed on 27 January 2023), and, for the SLCO1B1 transporter, the list from Karlgren et al. (2012) [25]. 

### 2.4. Identification of Gene Variants Potentially Enhancing the Effect of DDI Perpetrators

To identify the gene variants that could potentiate the effect of DDI perpetrators, we looked for gene variants of CYP3A4/5, CYP2C9, CYP2C19, CYP2D6 and SLC1B1 in some of the major pharmacogenomic databases including PharmVar (https://www.pharmvar.org/, accessed on 14 February 2023), PharmGKB (https://www.pharmgkb.org, accessed on 16 February 2023), dbSNP (https://www.ncbi.nlm.nih.gov/snp/, accessed on 17 February 2023), ensembl (https://www.ensembl.org/index.html, accessed on 14 February 2023) and 1000 genomes (https://www.internationalgenome.org/, accessed on 18 February 2023). Moreover, we retrieved from these databases information on the prevalence of these variants in Europe, in Italy, and whenever available, in Southern Italy. To investigate the possible existence of additional information on gene variant prevalence not represented in the above-mentioned databases, we also performed a systematic search on PubMed (https://pubmed.ncbi.nlm.nih.gov/, accessed on 27 February 2023) using as key words Italy, Southern Italy, South Italy and the various genetic variants of interest identified with either the rs dbSNP nomenclature or, when available, the star (*) allele designation.

### 2.5. Statistical Analysis

Statistical analysis of the data was performed using the IBM SPSS Statistic 26 software (1 New Orchard Road Armonk, New York, NY, USA). All data are reported as median and interquartile range. Data comparisons were performed with Wilcoxon signed-rank test. Prevalence data were compared using the χ^2^ test. The threshold for statistical significance was set at *p* < 0.05.

## 3. Results

### 3.1. Identification of the Most Prescribed Drugs in the Study Population

The study population consisted of 290 older adults (median age 74, IQR 69–79; 126 females): 221 (96 females) were admitted to the Internal Medicine ward and 69 (30 females) were followed as outpatients at the Geriatric Clinics.

All patients were affected by multiple comorbidities, the most frequent of which were arterial hypertension, dyslipidemia, type II diabetes, carotid artery atherosclerosis, ischemic heart disease (IHD), chronic renal failure, and chronic obstructive pulmonary disease (COPD) (Table 1). Most of the comorbidities showed a similar prevalence in the two genders with the only exceptions of arterial hypertension, hepatic cirrhosis, diverticulosis, osteoarthritis, osteoporosis, anxiety, and depression, which occurred more frequently in females, and IHD, peripheral artery disease and COPD, which were more prevalent in males. All patients were on polypharmacy and the average number of drugs was 8 (IQR 6–10).

Table 2 reports the list of the drugs prescribed to more than 5% of patients of our population. Most of them were either cardiovascular drugs, proton pump inhibitors (PPIs), or antidiabetic drugs. The list also included allopurinol, a drug for hyperuricemia, tiotropium an anticholinergic drug for COPD, rifaximin, an antibiotic frequently prescribed for colonic diverticula, and tamsulosin, an alpha-adrenergic blocker for prostate hyperplasia. No significant difference was observed between males and females in the prevalence of use of the different drugs with the only exceptions of hydrochlorothiazide, potassium canrenoate and enoxaparin, which were more frequently prescribed in females, and atorvastatin, low-dose aspirin, clopidogrel, carvedilol, and spironolactone, which were more prevalent in males.

### 3.2. Identification of Gene Polymorphism Potentially Enhancing the Effect of Perpetrator Drugs

Table 3 lists the inhibitors and inducers of CYPs and SLC1B1 identified among the drugs most prescribed in older adults and summarizes the main findings of our search in genomic databases for gene variants which could be responsible for DDGIs. We found seven different active principles that may act as drug perpetrators in DDGIs involving CYP3A4: one of them, amiodarone, is a strong inhibitor of this cytochrome, four are weak inhibitors (amlodipine, esomeprazole, omeprazole and pantoprazole), one is a strong inducer (rifaximin) and one is a weak inducer (warfarin). Genomic database analysis showed that in the European population the most represented loss of function variant, potentially involved in those DDGIs, are CYP3A4*1G [26,27,28] and CYP3A4*22 [29] which occur, respectively, in about 8% and 5% of the European population (https://www.internationalgenome.org/, accessed on 18 February 2023).The in/del CYP3A4*20 allele [30,31,32] also causes a complete loss of function, but it is observed only in about 0.04% of the European population (https://gnomad.broadinstitute.org/, accessed on 20 February 2023) (Table 3). Data on the prevalence of CYP3A4 alleles in Italy were retrieved from the *Tuscans in Italy* cohort of the 1000 genome project, which reports a prevalence of 8% and 7.5% for CYP3A4*1G and CYP3A4*22, respectively, and from Apellániz-Ruiz et al. (2015) [33], who showed the absence in the Italian population of the CYP3A4*20 allele. A small series of 50 epileptic patients from Southern Italy reports a prevalence of 6% of CYP3A4*22 in heterozygosity (*1/*22) [34]. A gain of function CYP3A4 variant, CYP3A4*18, has been described in Asians [35], but it is apparently absent in the European population (https://www.ensembl.org/, accessed on 14 February 2023). An additional factor that might impact on the DDGI involving CYP3A4 is the genetic variability of CYP3A5, a polymorphic cytochrome closely related to CYP3A4 and sharing most of its substrates [36]. Database analysis showed that in the European population, the most represented variant of CYP3A5 is the loss of function CYP3A5*3 allele with a prevalence of 92.4% whereas the normal allele CYP3A5*1 occurs only in 7.4% of the population (the 1000 genomes project, https://www.internationalgenome.org/, accessed on 18 February 2023). Data from the *Tuscans in Italy* cohort showed that the prevalence of CYP3A5*1 is low also in the Italian population and we found similar results also in several small series from Central and Northern Italy [37,38,39]. We found only one published study on CYP3A5 alleles in Southern Italy reporting a prevalence of only 1.3% for the *1/*1 fully functional diplotype [40]. Amiodarone is the only CYP2C9 (moderate) inhibitor in the list of the drugs most prescribed in our older adult cohort, which also includes two inducers of this cytochrome, one of moderate (rifampicin) and the other of weak potency (warfarin). In the European population, the most represented CYP2C9 loss of function gene variant is CYP2C9*3 with a prevalence of 7.3%, whereas CYP2C9*2 is the most frequently occurring intermediate activity allele, showing a prevalence of 12.4% (https://www.internationalgenome.org/, accessed on 18 February 2023). The few studies that assessed the prevalence of CYP2C9 alleles in Italy report values ranging from 8.4 to 16.7% for the loss of activity CYP2C9*3 allele, and from 12.4 to 35.6% for the intermediate activity CYP2C9*2 allele (https://www.internationalgenome.org/, accessed on 18 February 2023) [41,42,43,44], with no significant regional difference between Northern, Central and Southern Italy [45]. Five of the active principles more frequently prescribed to older adults might act as drug perpetrators in DDGIs involving CYP2C19, four weak inhibitors (amiodarone, esomeprazole, omeprazole and pantoprazole) and one strong inducer, rifaximin. Database analysis showed that CYP2C19*2, CYP2C19*8, CYP2C19*4, and CYP2C19*3 are the loss of function alleles of this highly variable cytochrome [46] that occur more frequently in Europe (prevalence: 14.5%, 0.3%, 0.20% and 0.17%, respectively). CYP2C19 hyperfunctioning variants are also highly represented [47]; among these, the CYP2C19*17 allele, which could theoretically enhance the effect of the inducers of this cytochrome, ranks second for prevalence (21.6%) in Europe among CYP2C19 alleles. Significant differences in the regional prevalence of CYP2C19 variants have been observed among the Italian macroregions; in fact, poor metabolizer variants with low enzyme activity were not observed in Southern Italy but occurred in about 5% and 1.6% of people from Central and Northern Italy, respectively [45]. Three weak CYP2D6 inhibitors, amiodarone, amlodipine, and omeprazole, appear in the list of the drugs most frequently used in our cohort of geriatric patients. Genetic database search showed that loss of function variants of this cytochrome are highly represented in the European population with prevalence of 18.5, 2.95, 1.59, and 1.11%, respectively, for the CYP2D6*4, CYP2D6*5, CYP2D6*3 and CYP2D6*6 alleles (https://www.pharmgkb.org/, accessed on 16 February 2023). In addition, some intermediate activity variants occur frequently in Europe, including CYP2D6*41, CYP2D6*10, CYP2D6*9 and CYP2D6*17 with a prevalence of 9.4, 1.57, 2.76 and 0.39%, respectively. Based on the data from the *Tuscans in Italy* cohort and from several additional small series [43,48,49], these loss of function and intermediate function variants seem to also be highly prevalent in Italy. Marked differences have been observed between Northern Italy, where loss of function alleles are more represented, and Central and Southern Italy, where the prevalence of intermediate variants is higher [45].

Five SLCO1B1 inhibitors (Atorvastatin, Digoxin, Pantoprazole, Rosuvastatin, Simvastatin) appear in the list of the drugs most frequently prescribed to the patients of our cohort. The effect of these inhibitors is expected to be potentiated in the presence of SLCO1B1 variants with reduced or null activity. In the European population, the nonfunctioning variants SLCO1B1*5 and SLCO1B1*15 account for about 17% (2.04 and 15.02, respectively) of all SLCO1B1 alleles. The nonfunctioning SLCO1B1 variants *5, *15 and *17 contain all the rs4149056 C SNP, which is, therefore, often used to identify the loss of function of this transporter. In the *Tuscans in Italy* cohort (https://www.internationalgenome.org/, accessed on 18 February 2023), the prevalence of rs4149056 C is 21.5%.

### 3.3. Currently Available CDSS with Pharmacogenomic Integration in DDIs Checkers

We searched the web and the current scientific literature to identify CDSS that include an evaluation of pharmacogenomic-related factors and to assess whether any of them take into account the role of these variations in determining the risk of DDIs. Table 4 reports the main currently available pharmacogenomic-based CDSS that we identified in our search. Most of them have been developed by University Hospitals for internal use upon integration with their local electronic health records (EHRs). These systems have been usually designed as stand-alone tools to suggest dose adjustments or changes of therapy in patients with specific gene variants (e.g., for thiopurines and thiopurine-methyltransferases) and not as more complex systems that incorporate pharmacogenomic information into larger databases of DDGIs. This is the case, for instance, of the CDSS developed at the Clinical Pharmacogenomics Service of the Boston Children’s Hospital (https://www.childrenshospital.org/centers-and-services/programs/a-_-e/clinical-pharmacogenomics-service-program#, accessed on 12 April 2023), of the Genomic Prescribing System (GPS) of the Center for Personalized Therapeutics, University of Chicago (https://cpt.uchicago.edu/gps/, accessed on 12 April 2023), of the CDSS developed at the University of Washington, Seattle to provide PGx-related alerts in the fields of oncology and cardiology [50] and FARMAPRICE, a prototype PGx-based CDSS, which is intended for a larger scale implementation but is currently tested at the Italian Centro di Riferimento Oncologico -Aviano Hospital mainly for oncological patients [51]. We found only a few CDSS which also analyze DDIs and DDGIs. The YouScript Precision Prescribing Software (https://youscript.com/what-we-do/clinical-decision-support-software/, accessed on 10 April 2023) is a fully developed, commercially available platform that incorporates testing for DDIs, DGIs and DDGIs. It includes data on more than 4000 drugs, herbal remedies, and OTC drugs. YouScript is fully compatible with EHRs, and a mobile device version is also available. GenXys (https://www.genxys.com/, accessed on 10 April 2023) is a software solution package for precision medicine, which includes two software tools: the TreatGx software, a CDSS for Pharmacogenetic Testing Interpretation and Precision Prescribing through the assessment of potential DGIs, DDIs and DDGIs, and the ReviewGx (https://www.genxys.com/content/mtm-software/, accessed on 10 April 2023), a computer engine for PGx-based automated medication review. GenXys can be easily integrated with into EHRs, electronic medical records (EMRs), and Pharmacy Management Systems (PMS). Importantly, none of the examined CDSS is specifically designed to be used in geriatric medicine.

## 4. Discussion

In the present manuscript, we identified the most frequently prescribed perpetrator drugs in a cohort of older adults from Sothern Italy and searched the literature to find the gene variants enhancing their perpetrator effect, which are most prevalent in Europe, Italy and, more specifically Southern Italy. In particular, we focused on perpetrator drugs prescribed in at least 5% of our cohort of older adults which may be responsible for pharmacokinetics DDIs involving CYP3A4/5, CYP2C9, CYP2C19 and CYP2D6 and SLC1B1 transporters. These frequently prescribed drugs were mainly represented by antiplatelet drugs, statins and cardiovascular drugs including Angiotensin-converting-enzyme inhibitors (ACE-I), β-blockers and diuretics. Quite surprisingly, about 13% of the patients in our cohort took warfarin, not only because some of them were bearing mechanical valve prostheses but also because, in our retrospective series, most of them were admitted to the hospital before Direct Oral Anti-Coagulant (DOAC) became the standard for oral anticoagulation. Three PPIs, omeprazole, esomeprazole, and pantoprazole, were also included in the list of drugs prescribed in at least 5% of our patients; although the largest fraction of patients assumed PPIs together with low-dose aspirin, these drugs were also prescribed in association with other antiplatelet drugs or without any antithrombotic treatment, as also reported in other published studies [79,80,81]. Allopurinol ranked eighth in our list, being prescribed in about 18% of the older adults of our cohort even though none of them was affected with gout, and the use of allopurinol for secondary hyperuricemia is not recommended by current guidelines [82,83]. Quite unexpectedly, no psychiatric drug appeared among those prescribed the most. This result could be explained both by the source of our data that were mostly collected in an internal medicine ward with a strong cardiovascular specialization and by the fact that, in our database, several different drugs are used for similar indications (e.g., insomnia or agitation), thus leading to a “fragmented” prescription, with none of them reaching the 5% threshold.

Only a few of the drugs prescribed to more than 5% of our patients are potential DDI perpetrators, suggesting that it should be easy to spot them during the prescription process and to prevent or minimize their potential interactions. Among them, amlodipine is an inhibitor of CYP3A4 and CYP2D6, atorvastatin and digoxin block the SLCO1B1 transporter, and the PPIs esomeprazole, omeprazole and pantoprazole may inhibit CYP3A4/5, CYP2C19, CYP2D6 and SLCO1B1. It is also worth noting that amiodarone not only inhibits CYP3A4/5, CYP2C9, CYP219 and CYP2D6 but it also blocks the plasmamembrane pump ABC1B1 and, therefore, it may be responsible for clinically relevant drug interactions [84]. The only CYP inducers were warfarin, which induces the expression of CYP3A4/5 and CYP2C9, and rifaximin, an inducer of CYP3A4/5, CYP2C9 and CYP2C19, whose involvement in DDIs is, however, debatable and probably only minor considering its limited oral bioavailability.

In searching for genetic variations in CYPs and SLCO1B1 which could increase the effect of perpetrator drugs and be involved in DDGIs, we reasoned that variants reducing the activity or expression of these proteins should potentiate the effects of drug inhibitors whereas gain of function variants should potentiate the effect of drug inducers. Our search showed that several functionally relevant gene variants in cytochromes and SLCO1B1 occur in the European population with a non-negligible prevalence higher than 1% and sometimes up to 5%. They include, CYP3A4*22 and CYP3A5*1, for CYP3A4, CYP2C9*2 and CYP2C9*3 for CYP2C9, CYP2C19*17 and CYP2C19*2 for CYP2C19, CYP2D6*4, CYP2D6*5, CYP2D6*9, CYP2D6*10, and CYP2D6*41 for CYP2D6, and SLCO1B1*5 and SLCO1B1*15 for SLCO1B1. Previous studies largely demonstrated the ability of these variants to affect the pharmacokinetics of major drug classes including, for instance, clopidogrel, statins, immunosuppressant drugs, NSAIDs, warfarin and antiepileptic drugs even when given alone and independently from DDIs; accordingly, guideline recommendations and software solutions for dose correction in people with these variants have been developed [85,86,87,88,89,90,91].

On the contrary, only few studies investigated their relevance in DDGIs. In the seminal paper by Verbeurgt et al. (2014) [14], drug therapy was examined with the YouScript^®^ CDSS for DDIs, DGIs and DDGIs in 1143 patients aged 18–89 years. Potential DDGIs were detected in 12% of the study population and the drugs more frequently involved in interactions were metoprolol, clopidogrel, simvastatin, aspirin and hydrocodone. The YouScript^®^ CDSS was also used in the study by Hocum et al. (2016), [22] in which drug prescriptions from 22,000 patients (age range, 1–108 years with a mean of 60 years) were examined for DDI, DGI and DDGI occurrence. Patients were stratified for age, higher or lower than 65, and results showed that DGIs and DDGIs requiring changes in therapy were 120% more frequent in older adults. One hundred older adults were examined by Bain et al. (2019) [21] as part of the Program of All-inclusive Care for the Elderly (PACE) population. On average, three DDGIs were identified per participant with the help of a proprietary CDSS (Medication Risk Mitigation™ Matrix, CareKinesis, NJ, USA) and more than one-third of them involved CYP2D6 variants. The perpetrator drugs responsible for most of these DDGIs were Metoprolol, Pantoprazole, Oxycodone, Trazodone, Duloxetine, Hydrocodone, Sertraline and Tamsulosin. This list of major interacting drugs differs from what we found in our study for the presence of CNS drugs and tamsulosin. Among the factors possibly responsible for these differences is the fact that our study population was mainly represented by patients from internal medicine wards/consultation and that the prescription of antidepressant drugs was highly fragmented with a large number of different pharmaceutical ingredients used for this purpose, none of which reached the 5% threshold set in our study.

The present study did not directly identify the victims of the potential DDGIs that could be caused by the combos between perpetrator drugs and pharmacogene variants. However, several theoretical predictions may be formulated and are summarized in Table 5. More specifically, CYP3A4 metabolizes many drugs frequently prescribed in older adults, such as amiodarone, amlodipine, buprenorphine, citalopram, diltiazem, manidipine, ranolazine, verapamil and many statins and PPIs [17]. Clopidogrel, which is used either alone or in combination with low-dose aspirin for the secondary prevention of cardiovascular events, PPIs, and SSRI are among the substrates of CYP2C19 more frequently prescribed to older adults, whereas CYP2D6 participates to the metabolism of β-blockers, antidepressants belonging to the families of Selective Serotonin Reuptake Inhibitors (SSRIs) and Serotonin and Noradrenaline Reuptake Inhibitors (SNRIs), antipsychotics, such as clozapine and quetiapine, the analgesics codeine and tramadol, and the anticancer drug tamoxifen [92]. Real world data are needed to substantiate these theoretical predictions; indeed, we are currently performing at our institution a prospective study aiming to identify the victim drugs that are most commonly associated in older adults with the gene–drug combos that we identified.

The results of the present study suggest that the characterization of the genetic patterns potentiating the effects of drug perpetrators could help identify geriatric patients at high risk of dangerous drug interactions. With information about the pharmacogenotype of their patients in mind, physicians could avoid dangerous drugs, choosing safer alternatives or correcting their dosage accordingly to genotype and to the potentially coprescribed drug perpetrators. It is important, however, to emphasize that for the routine implementation of pharmacogenomics both in the specific context of geriatrics and, more in general, in clinical medicine, solid evidence about its efficacy and cost-effectiveness is still awaited and the development of a larger set of detailed guidelines will be needed as well [93]. An additional potential advantage of pharmacogenotyping is that it could enable physicians to correctly interpret the warnings provided by medication review reports and CDSS, hence, avoiding alert fatigue. Considering that only a limited number of potentially relevant gene variants occurs in the European population, the inclusion of pharmacogenetic testing in the diagnostic workup that precedes drug prescription in older adults should be practically feasible and economically sustainable. The result of this testing should be included among the parameters evaluated by CDSS to identify DDGIs. This is a field that needs to be further developed also with tools more specifically designed for older adults. In fact, our web search showed that very few pharmacogenomic-oriented CDSS with DDGI checkers are available. Some of them have been successfully used to detect potential DDGIs [14,22] but none of them has been specifically designed for older adults. An important practical problem in designing such a kind of CDSS is represented by the need to identify active principles from drug brand names, which often differ from one country to the other. In this perspective, the efforts of developing a univocal identification of medicines as foreseen through the adoption of IDMP codes appear of major relevance also considering that they could be modified to include key pharmacogenomic information among critical drug attributes [94].

The present study has several points of strength and limitations. The strength of the present study is that we focused on a cohort of patients from a small geographic area. The prevalence of variations in specific pharmacogenes may, indeed, significantly change from one region to another, as has been demonstrated, for instance, for several CYPs in Italy subregions by Carano et al. (2014) [45]. This suggests that geographic and ethnicity factors should be considered in the evaluation of patient risk for DDGIs. It has, however, to be considered that because of massive, unprecedented migratory fluxes, the ethnicity of the European population is rapidly changing. This implies that an intrinsic limitation of our study is that the prevalence figures of specific pharmacogene variants that we retrieved from genomic databases could not accurately reflect the actual prevalence of these variants whose assessment would require constantly updated data. A second limitation of our study is that a significant percentage of our study population included older adults admitted to the Internal Medicine ward of the Federico II University hospital because of cardiovascular disorders, and this has likely biased the estimated prevalence of prescribed drugs toward cardiovascular, antiplatelet, anticoagulant and lipid-lowering drugs. An additional limitation is that we evaluated single active principles individually and did not consider drug classes (or subclasses) as a whole. While this was a forced choice since in many cases different members of the same class have different pharmacokinetics, it also led to a gross underestimation of the impact of drug classes, which have many members and whose prescription is, therefore, highly fragmented. For instance, this might have been the case of SSRI. Finally, a further limitation of the present study is that our predictions about the gene variants potentially relevant for DDGIs were based only on the search on already available data from databases or scientific papers and not on the genotyping of our patients. Such an approach has, however, the advantage of providing prevalence estimates on a larger scale than the small regional scale given by patients recruited at a single institution. Finally, in our analysis we did not consider pharmacodynamic interactions, which are also well-known to represent an important cause of drug toxicity.

## 5. Conclusions

In conclusion, the present study provides a list of the most prescribed drugs potentially acting as perpetrator drugs in older adults and of the allelic variants, which could enhance their effects in a significant percentage of European older adults. This information might be instrumental for designing pharmacogenomics oriented CDSSs and, ultimately, to optimize the medicine review process.

## Figures and Tables

**Table 1 geriatrics-08-00084-t001:** Main demographic and clinical characteristics of the study population.

Disease	Whole Population (n = 290)	Males (n = 164)	Females (n = 126)
Age	74 (69–79)	73 (68–78)	75 (70–80)
Prescribed drugs number	8 (6–10)	8 (6–10)	8 (7–10)
Comorbidities number	6 (4–8)	6 (4–7)	7 (5–9)
Arterial Hypertension	185 (63.8)	88 (53.7)	97 (77.0) ***
Type II DM	112 (38.6)	58 (35.4)	54 (42.9)
Dyslipidemia	101 (34.8)	56 (34.1)	45 (35.7)
Ischemic Heart Disease	90 (31.0)	68 (41.5)	22 (17.5) ***
Carotid artery atherosclerosis	81 (27.9)	42 (25.6)	39 (30.9)
CKD	71 (24.5)	38 (23.2)	33 (26.2)
Atrial fibrillation	66 (22.8)	41 (25.0)	25 (19.8)
COPD	58 (20.0)	40 (24.4)	18 (14.3) *
Benign Prostatic Hyperplasia	45 (15.5)	45 (27.5)	--
Goiter	34 (11.7)	14 (8.5)	20 (15.9)
Anemia	30 (10.3)	17 (10.4)	13 (10.3)
History of cancer	30 (10.3)	12 (7.3)	18 (14.3)
Diverticulosis	25 (8.6)	8 (4.9)	17 (13.5) **
Depression and anxiety	23 (7.9)	7 (4.3)	16 (12.7) **
Osteoarthritis	22 (7.6)	4 (2.4)	18 (14.3) ***
Peripheral artery disease	21 (7.2)	20 (12.2)	1 (0.8) ***
Chronic hepatitis	19 (6.6)	7 (4.3)	12 (9.5)
Hepatic cirrhosis	19 (6.6)	3 (1.8)	16 (12.7) ***
Osteoporosis	11 (3.8)	2 (1.2)	9 (7.1) **
GERD	11 (3.8)	4 (2.4)	7 (5.6)
Angina	10 (3.4)	4 (2.4)	6 (4.8)

Comorbidities are listed in order of prevalence. For each comorbidity we reported the number of patient and, in parentheses, the percentage in the respective group. *, *p* < 0.05; **, *p* < 0.01; ***, *p* < 0.001 at χ^2^ test. Abbreviations: CKD, Chronic Kidney Disease; COPD, chronic obstructive pulmonary disease; DM, Diabetes Mellitus; GERD, Gastroesophageal reflux disease.

**Table 2 geriatrics-08-00084-t002:** Active principles prescribed to 5% or more of the patients in study population.

	Whole Population	Males	Females
Low dose aspirin	124 (42.8)	85 (51.8)	39 (31.0) ***
Furosemide	104 (35.9)	62 (37.8)	42 (33.3)
Atorvastatin	104 (35.9)	68 (41.5)	36 (28.6) *
Esomeprazole	77 (26.6)	39 (23.8)	38 (30.2)
Pantoprazole	76 (26.2)	38 (23.2)	38 (30.2)
Clopidogrel	70 (24.1)	48 (29.3)	22 (17.5) *
Ramipril	69 (23.8)	44 (26.8)	25 (19.8)
Allopurinol	60 (20.7)	36 (22.0)	24 (19.0)
Carvedilol	58 (20.0)	40 (24.4)	18 (14.3) *
Amlodipine	55 (19.0)	30 (18.3)	25 (19.8)
Metformin	54 (18.6)	28 (17.1)	26 (20.6)
Omeprazole	54 (18.6)	33 (20.1)	21 (16.7)
Bisoprolol	52 (17.9)	24 (14.6)	28 (22.2)
Hydrochlorothiazide	49 (16.9)	19 (11.6)	30 (23.8) **
Insulin glargine	48 (16.6)	25 (15.2)	23 (18.3)
Warfarin	37 (12.8)	18 (11.0)	19 (15.1)
Potassium canrenoate	35 (12.1)	14 (8.5)	21 (16.7) *
Insulin Lispro	33 (11.4)	18 (11.0)	15 (11.9)
Digoxin	33 (11.4)	21 (12.8)	12 (9.5)
Tiotropium	31 (10.7)	19 (11.6)	12 (9.5)
Olmesartan	30 (10.3)	14 (8.5)	16 (12.7)
Irbesartan	29 (10.0)	14 (8.5)	15 (11.9)
Simvastatin	28 (9.7)	17 (10.4)	11 (8.7)
Spironolactone	26 (9.0)	22 (13.4)	4 (3.2) **
Nebivolol	22 (7.6)	11 (6.1)	11 (8.7)
Nitroglycerin	21 (7.2)	13 (7.9)	8 (6.3)
Rosuvastatin	21 (7.2)	10 (6.1)	11 (8.7)
Folic Acid	20 (6.9)	13 (7.9)	7 (5.6)
Tamsulosin	19 (6.6)	19 (11.6)	--
Insulin aspart	19 (6.6)	9 (5.5)	10 (7.9)
Doxazosin	18 (6.2)	8 (4.5)	10 (7.9)
Rifaximin	18 (6.2)	10 (6.1)	8 (6.3)
Ursodeoxycholic acid	17 (5.9)	7 (4.3)	10 (7.9)
Amiodarone	17 (5.9)	14 (8.5)	3 (2.4)
Atenolol	16 (5.5)	9 (5.5)	7 (5.6)
Enoxaparin	16 (5.5)	5 (3.0)	13 (10.3) *
Telmisartan	15 (5.2)	6 (3.7)	9 (7.1)

Active pharmaceutical ingredients are listed in order of prevalence. For each active principle we reported the number of patients and, in parentheses, the percentage in the respective group. *, *p* < 0.05; **, *p* < 0.01; ***, *p* < 0.001 at χ^2^ test.

**Table 3 geriatrics-08-00084-t003:** Inhibitors and inducers of CYPs and SLCO1B1 and main gene variants potentially enhancing their effects on drug disposition.

	Inhibitors	Inducers	Main Gene Variants	Functional Effect	Prevalence in Europe (1)	Prevalence in Italy	Prevalence in Southern Italy
CYP3A4/5	amiodarone, amlodipine, esomeprazole, omeprazole, pantoprazole	warfarin, rifaximin	CYP3A4*22	LoF	5%	3.7% (2)	3% (3)
			CYP3A4*1G	IM	8%	8.4 (2)	N/A
			CYP3A5*3	LoF	92.4%	94.9% (2)	96.6 (4)
CYP2C9	amiodarone	warfarin, rifaximin	CYP2C9*2	IM	12.4%	17.6% (5)	13.6% (5)
			CYP2C9*3	LoF	7.3%	9.5% (5)	10.0% (5)
CYP2C19	amiodarone, esomeprazole, omeprazole, pantoprazole	rifaximin	CYP2C19*17	GoF	21.6%	17.6 (5)	N/A
			CYP2C19*2	LoF	14.5%	13.8% (5)	6.4% (5)
			CYP2C19*8	LoF	0.3%	0 (2)	N/A
CYP2D6	amiodarone, amlodipine, omeprazole	----	CYP2D6*4	LoF	18.5%	14.9% (5)	11.82 (5)
			CYP2D6*41	IM	9.2%	15.2% (5)	18.2 (5)
			CYP2D6*5	LoF	2.95%	0.9% (5)	0.9 (5)
			CYP2D6*10	IM	1.6%	2.6% (5)	3.6 (5)
			CYP2D6*9	IM	2.8%	1.7% (5)	0.9 (5)
			CYP2D6*17	IM	0.4%	0.3% (5)	0 (5)
SLCO1B1	atorvastatin, digoxin, pantoprazole, rosuvastatin, simvastatin	----	SLCO1B1*15	LoF	15.0%	N/A	N/A
			SLCO1B1*5	LoF	2.0%	N/A	N/A
			rs4149056	LoF	16.1%	21.5% (2)	N/A

(1) Prevalence data in Europe were obtained from the ensembl database (https://www.ensembl.org/, accessed on 14 February 2023) and from the PharmGKB PGx Gene-specific Information Tables (https://www.pharmgkb.org/page/pgxGeneRef, accessed on 16 February 2023). (2) Data from the Tuscans in Italy cohort of the 1000 genomes phase 3 (https://www.internationalgenome.org/, accessed on 14 February 2023) as reported in the ensembl database (https://www.ensembl.org/, accessed on 14 February 2023). (3) Caruso et al., 2014 [34]. (4) Provenzani et al., 2011 [40]. (5) Carano et al., 2017 [45]. Abbreviations: GoF: gain of function; IM: intermediate function; LoF: loss of function; N/A: not available.

**Table 4 geriatrics-08-00084-t004:** CDSS including pharmacogenetic factor evaluation.

Name of the System or of the Institution/Project	Main Features	DDGI	Ref.
Clinical Pharmacogenomic Service/Boston Children’s Hospital	A software solution developed for internal use at the Boston Children’s Hospital Clinical; fully integrated with the EHR it generates alerts based on PGx-testing results upon drug prescribing.	NO	[52]
University of Washington, Seattle	A prototype developed at the University of Washington, Seattle to incorporate into the PowerChart^®^/Cerner Millennium^®^ environment, a semi-active PGx-based CDSS which upon prescription of selected drugs triggers either an alert for ordering PGx testing, or, when PGx data are already available, or displays a link to e-resources to provide information to support clinical decision.	NO	[50,53,54,55,56]
RIGHT/Mayo Clinic https://www.mayo.edu/research/clinical-trials/cls-20316196 (accessed on 27 January 2023)	A CDSS developed at Mayo Clinic for internal use as a tool of the *Right Drug*, *Right Dose*, *Right Time* project on preemptive PGx testing in precision medicine. The system generates PGx alerts at the time of drug prescription by interacting with a EHR in which data on preemptive genotyping of 85 pharmacogenes are stored.	NO	[57,58,59]
PREDICT/Vanderbilt University Medical Center https://www.vumc.org/predict-pdx/welcome (accessed on 27 January 2023)	A locally developed EHR supporting the request for PGx testing either preempting or upon prescription of specific drugs. The system stores genomic data until drugs that could generate DGIs are prescribed; at that time PGx-related alert, a list of potential DGIs and advice for therapy adjustments are generated.	NO	[60,61]
Personalized Medication Program/University of Florida	An EHR modified for the preemptive request of CYP2C19 for patients undergoing cardiovascular procedures at the University of Florida. After storage of patient PGx data the system automatically generate a BPA (best practice advice) whenever a CYP2C19 drug substrate is prescribed.	NO	[62]
Personalized Medication Program/Cleveland Clinic Health System	A PGx software developed at the Cleveland Clinic as a complement of the My Family prescription tool, which reports family health information in the HER; it prompts clinicians to ordering PGx testing when prescribing selected drugs or, if this information is already available, it displays PGx results together with BPAs (best practice advices) for PGx-guided drug prescription.	NO	[63]
PG4KDS/St. Jude Children Research Hospital https://www.stjude.org/treatment/clinical-trials/pg4kds-pharmaceutical-science.html (accessed on 10 April 2023)	An automated system developed at the St. Jude Children Research Hospital as part of the PG4KDS project to incorporate into the EHR the results of preemptive testing of 225 pharmacogenes, their clinical interpretation and, when available, direction on drug prescription and dose adjustments according to CPIC guidelines.	NO	[64,65,66]
CLIPMERGE PGx/The Mount Sinai Hospital	A PGx knowledge platform independent from, but fully integrated with the Mount Sinai’s Epic HER; it has been developed to generate alerts, and suggest specific corrective actions upon drug prescription based on the drug prescribed and the results of patient genetic testing.	NO	[67]
FARMAPRICE/Centro Oncologico di Aviano	A prototype PGx-based CDSS to identify potential DGIs and suggest therapy adjustment developed at the Centro Oncologico di Aviano and currently tested mainly on oncological patients.	NO	[51]
GPS/University of Chicago https://cpt.uchicago.edu/gps/ (accessed on 12 April 2023)	A web-based portal developed by the Center for Personalized Therapeutics of the University of Chicago to support PGx-based drug prescription at the Chicago University Medical Center.	NO	[68,69,70]
Medication Safety Code (MSC)/University of Vienna https://safety-code.org/ (accessed on 14 April 2023)	A research prototype service available upon request that generates a QR code containing the results of patient genetic testing. This QR code is printed onto a plastic card and after scanning provides web-based patient-specific dosing recommendations.	NO	[71,72]
GIMS (Genetic Information Management Suite/the U-PGx project) https://upgx.eu/ (accessed on 14 April 2023)	A knowledge database developed in the context of the UPGx project to support the implementation of PGx-based drug therapy adjustments in the CDSS already available at the clinical sites participating to the project.	NO	[73]
GeneSight https://genesight.com/ (accessed on 16 April 2023)	A commercial service that performs genetic testing for patients who have to be given psychotropic drugs and also provides a short report with PGx-oriented recommendations for drug prescription.	NO	[74]
YouScript https://youscript.com/ (accessed on 10 April 2023)	A commercial CDSS software solution for the combined evaluation of DGIs and DDGIs. It covers not only prescription drugs but also herbal remedies and OTC medicine. Full integration with EHR.	YES	[19,75]
GenXys https://www.genxys.com/content/ (accessed on 10 April 2023)	A commercial software suite which also includes a tool for precision prescribing based on PGx testing results (TreatGx) and a software for automated medicine review (ReviewGx) which also includes PGx-based recommendations and advice for drug deprescribing.	YES	[76]

Data obtained from web search and Blagec et al. (2018) [72], Hinderer et al. (2017) [77], and Roosan et al. (2020) [78].

**Table 5 geriatrics-08-00084-t005:** Some notable examples of Drug Perpetrator–Gene Combinations and of their main potential victims and clinical implications.

Drug Perpetrator–Gene Combination	Main Victims	Potential Clinical Consequences of DDGIs
CYP3A4/5 LoF or IM variants + CYP3A4/5 inhibitors	amlodipine, diltiazem, verapamil	Hypotension, bradyarrhytmias
	atorvastatin, simvastatin	Increased risk of myopathy
	quetiapine	Increased in drug toxicity (e.g., hypotension, dizziness, drowsiness, QT prolongation, hyperlipidemia, hyperglycemia), loss of antidepressant activity
	tacrolimus	Increased in drug toxicity (e.g., opportunistic infections, hyperglycemia, hyperlipidemia, hypertension, nephrotoxicity, hepatotoxicity)
CYP2C9 LoF or IM variants + CYP2C9 inhibitors	Phenytoin	Ataxia, dizziness, drowsiness, nystagmus, hepatotoxicity, megaloblastic anemia, leukopenia, hepatotoxicity, osteoporosis
	celecoxib, ibuprofen, flurbiprofen, meloxicam	Diarrhea, dyspepsia, vomiting, heartburn, increased risk of peptic ulcer and gastric bleeding,
	Fluvastatin	Higher myopathy risk
	Warfarin	Increased risk of bleeding
CYP2C19 LoF or IM variants + CYP2C19 inhibitors	Clopidogrel	Loss of clopidogrel efficacy: increased risk of ischemic cardiovascular disease
	omeprazole, lansoprazole, pantoprazole, dexlansoprazole	Increased risk of bone fractures, of gastrointestinal and respiratory tract infections, of vitamin and electrolyte deficiencies, especially hypomagnesemia
	SSRI (citalopram, escitalopram, sertraline)	Headache, drowsiness, blurred vision, tremor, xerostomia, nausea, vomiting, increased risk of falls, of SIADH, and of serotonin syndrome
	Voriconazole	Central neurotoxicity (confusion, hallucinations), hepatotoxicity
CYP2D6 LoF or IM variants + CYP2D6 inhibitors	SSRI (paroxetine, fluvoxamine)	Headache, drowsiness, blurred vision, tremor, xerostomia, nausea, vomiting, increased risk of falls, of SIADH, and of serotonin syndrome
	SNRI	Tachycardia, hypertension, mydriasis, insomnia, xerostomia, nausea, vomiting, increased risk of falls, of SIADH, and of serotonin syndrome
	Codeine, tramadol	Loss of codeine and tramadol efficacy: uncontrolled pain
	β-blockers (metoprolol)	Severe bradycardia
	Tamoxifen	Loss of tamoxifen efficacy
SLCO1B1 LoF variants + SLCO1B1 inhibitors	atorvastatin, rosuvastatin, simvastatin	higher myopathy risk
	Enalaprilat, olmesartan, valsartan	cough

Abbreviations: IM: intermediate function LoF: loss of function.

## Data Availability

The data presented in this study are available on request from the corresponding author. The data are not publicly available due to privacy restrictions.

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
