# Peer review of "Identification of Drugs Acting as Perpetrators in Common Drug Interactions in a Cohort of Geriatric Patients from Southern Italy and Analysis of the Gene Polymorphisms That Affect Their Interacting Potential"

_geriatrics, 2023, doi:10.3390/geriatrics8050084_

Round 1
Reviewer 1 Report
This manuscript is showing that Drug-Drug Interactions can be relevant to decide how to treat the older adults; it also provides some hints and clinical decision support systems to optimize geriatric medicine. Strenght and limits are definitively described.
Author Response
Reply to reviewer 1
Many thanks for your positive comments.
Reviewer 2 Report
Authors evaluated a list of the most prescribed drugs potentially acting as perpetrator drugs in older adults and the effect of allelic variants. This research provides inspiration for designing pharmacogenomics oriented CDSSs. Overall, the manuscript reads well and with some minor comments mentioned below:
1. In the introduction section, it would be better to add the relationship between DDIs and CYP3A4/5, CYP2C9, CYP2C19, CYP2D6 and SLC1B1. Since, line 120 shows this research focused on these cytochromes and transporters and they are responsible for most of the DDIs. This information should be written in more detail in the introduction.
2. It would be better to provide information on adverse effect or gene variants in the study population. This will help readers understand the relationship between gene variation and DDIs.
The manuscript is nicely written and reads well.
Author Response
Reply to reviewer 2
In the Introduction section, it would be better to add the relationship between DDIs and CYP3A4/5, CYP2C9, CYP2C19, CYP2D6 and SLC1B1. Since, line 120 shows this research focused on these cytochromes and transporters and they are responsible for most of the DDIs. This information should be written in more detail in the introduction.
Thanks for your suggestion: we agree that since the manuscript deals about CYPs and SLCO1B1, they should be mentioned in the Introduction. To fulfil your request, in the introduction of the new version of the we introduced a few sentences on these enzymes and transporters clarifying their major pharmacogenomic implications (page 2, lines, 75-96). We also added a new reference on SLCO1B1 (new refs. 18), and we moved to the Introduction the reference on CYPs by Zanger and Schwab (now ref 17).
It would be better to provide information on adverse effect or gene variants in the study population. This will help readers understand the relationship between gene variation and DDIs.
We agree that providing information on adverse effects and gene variants in our population would help the readers understand the relevance of DDGIs. Unfortunately, this was a retrospective study and such data are not available since they are not routinely collected in the patients followed at our institution.
Reviewer 3 Report
Authors have compiled well on Identification of drugs acting as perpetrators in common drug interactions in a cohort of geriatric patients from Southern Italy and analysis of the gene polymorphisms that can affect their interacting potential.
English is well.
Author Response
Reply to reviewer 3
Many thanks for your positive comments.
Reviewer 4 Report
The manuscipt of Cataldi and coll. identified drugs acting as perpetrators in DDI and gene polymorphisms potentially affecting their interaction.
The paper is clear and well-written.
Only one question: How would the patients recruited in the study benefit of the identification of gene variants enhancing/attenuating the effect of DDI perpetrators?
Author Response
Reply to reviewer 4
How would the patients recruited in the study benefit of the identification of gene variants enhancing/attenuating the effect of DDI perpetrators?…..
Thanks for raising the important question of the potential clinical impact of gene variants enhancing/attenuating the effect of DDI perpetrators. Prospectively, pharmacogenotyping could help reducing serious drug-drug interactions by preventively identifying risky patients in whom certain drug combinations should be avoided or major changes in drug dosage should be applied. This point was already partially addressed in the Discussion (page 13, lines 437-439 of the new version of the manuscript). To fulfil your request, we added two new sentences further emphasizing and, hopefully, clarifying this issue at page 13, lines 439-442 of the new version of the manuscript.
Reviewer 5 Report
Dear authors team, thank you for submitting an interesting paper with interesting methods utilizing a population approach. This paper highlights an important topic and the balance provided of the implications for clinical practice are helpful, including the review of clinical support tools that actually provide drug interaction information. Just a historical note, I kicked Genelex out of our clinic often for trying to get us to swab all of our geriatric patients about 7 years ago (they were incentivized by reimbursements to test) and they were even then trying to claim that PGx testing and their tool is the answer to all medication management. It is appreciated that your paper provides perspective on clinical utility in flagging patients to watch closely and keep drug interactions on our radar scope rather than focusing solely on PGx drug metabolism alone. Attached is a document with some feedback for your consideration. Good paper.

Author Response
Reply to reviewer 5
Problem with line 78-79: The suggestion that it is better to replace the concept of DDIs with that of drug-drug-gene interactions makes my mind uncomfortable…..
We would like to thank the reviewer for this comment which we totally agree with. To pinpoint that other factors besides the pharmacogenotype could affect the severity of specific drug interactions we added a new sentence (lines 99-105 of the new text) and we modified the sentence at lines 78-79 of the old manuscript (lines 105,107 of the new version of the manuscript).
Instead of the word “elderly” the term “older adult” should be substituted
As requested the term “elderly” has been replaced with the term “older adults” throughout the manuscript (lines 110, 342, 455 and 458) with the exception of line 402 where it is part of the acronym “Program of All-inclusive are for the Elderly (PACE)”.
Recommend caution in writing in the style that enforced assumptions that are not too clear just yet per current state of the art. Recommend softening statements throughout. For example lines 88-91 offer a more practical description of the role (and gaps) of PGx in care delivery
To address this important observation, we modified the sentence in the introduction at lines 107-111 of the new version of the manuscript. Moreover, we softened the sentence at lines 437-439 of the discussion and we added a new sentence on current limitations in pharmacogenomics in the discussion at page 13, lines 442-447 of the new version of the manuscript, which refers to the paper by Pirmohamed, mentioned by the reviewer.
There is a lot of great information in the discussion section. While reading, I found myself wanting to see a table or diagram that exemplifies the clinical utility of your findings
Thanks for the suggestion to add a table summarizing the potential implications of DDGIs which could greatly improve the readability of the Discussion. In the revised version of the manuscript we added such a table (table 5 at page 12 of the new version of the manuscript).
It might be of interest to mention the need for constant testing of at a population level due to the fast-changing European population (and around the world population) with unprecedented migration
Thank you for suggesting to give emphasis to this very important and up to date issue. To comply with your suggestion we added a new sentence in the Discussion (lines 470-475 of the new version of the manuscript).